# Diversity and Distribution of Macrofungi in Protected Mountain Forest Habitats in Serbia and Its Relation to Abiotic Factors

**DOI:** 10.3390/jof8101074

**Published:** 2022-10-13

**Authors:** Milana Rakić, Miroslav Marković, Zoran Galić, Vladislava Galović, Maja Karaman

**Affiliations:** 1Department of Biology and Ecology, Faculty of Sciences, University of Novi Sad, Trg Dositeja Obradovića 2, 21000 Novi Sad, Serbia; 2Institute of Lowland Forestry and Environment, University of Novi Sad, Antona Čehova 13, 21000 Novi Sad, Serbia

**Keywords:** macrofungi, diversity, distribution, forests, abiotic factors

## Abstract

Fungal diversity is one of the most important indicators of overall forest biodiversity and its health. However, scarce information exists on the state of macrofungal communities of mountain forests in Serbia, making it one of the countries with the least-published mycological data in the Mediterranean and Balkan region of Europe. This paper presents the results of the first comprehensive, long-term study of macrofungal communities in some of the most important mountain forest ecosystems in Serbia (Tara, Kopaonik and Vidlič)**.** In the course of three consecutive years, the sampling of five permanent experimental plots resulted in 245 species of macrofungi, classified into three functional groups (terricolous saprothrophs, lignicolous, and mycorrhizal fungi). Special attention was given to protected and indicator species, which point out the great value of studied forest habitats and the importance of their conservation. It was found that precipitation, habitat humidity, and temperature significantly influence the occurrence and distribution, primarily of mycorrhizal and lignicolous group of fungi. Thus, the continuation of long-term monitoring is crucial in order to more precisely determine which groups/species of macrofungi would, and to what extent they would, adapt to a rapidly changing climate.

## 1. Introduction

Forests are ecosystems with exceptional biodiversity. Nowadays, they are also among the most threatened ecosystems due to unsustainable management, pollution and climate change. Fungi play a very important role in the ecological balance of forest ecosystems. They represent crucial decomposers of organic matter, members of mycorrhizal communities necessary for the normal development of trees, as well as accumulators and degraders of harmful materials. Due to this, fungal diversity is one of the most important indicators of forests health [1]. Monitoring the state of fungal communities, their abundance, their dynamics of occurrence, and their interrelationships can serve as an early, extremely sensitive indicator of changes in a given environment (such as drought and eutrophication, soil acidification, pollution with harmful substances, and habitat changes in the form of deforestation) that can have a negative impact on plants species and especially on the trees that make up the main biomass of forest ecosystems [2,3,4].

Due to the exceptional ecological importance of macrofungi, research on their synecology, biogeography, and conservation status is becoming more and more plentiful [5,6,7,8,9,10,11,12,13]. Nevertheless, papers related to mycological research on forest habitats most often deal with the issue of only certain ecological groups of macrofungi: exclusively mycorrhizal [4,5,6,7,8,9,10,11,12,13,14,15,16,17,18] or lignicolous [8,19,20,21,22,23]. In contrast, studies covering the overall diversity of all ecological and taxonomic groups of macrofungi are considerably less frequent [24,25,26,27,28]. Thus, the importance of diverse data (concerning various aspects of mycopopulations) for understanding the vitality of the entire community of fungi and, consequently, forest habitats may be neglected.

Previous studies [29,30,31] indicate that increasingly frequent changes in climatic factors lead to specific changes within macrofungal communities, which relate to the dynamics of fructification, the number of recorded species, and the different contributions of individual functional groups (mycorrhizal, parasitic, and saprotrophic). The mentioned influences are also observed on a wider level, in terms of changes in the distribution range of individual species [32]. The literature data on the influence of abiotic factors on macrofungal diversity is very heterogeneous and inconsistent for several reasons. In the first place, there is a great variety of studied factors (various macro and microclimatic factors, edaphic factors); hence, it is difficult to perform a comparison with the literature data. The majority of studies address the effects of a single or just a few factors such as drought, precipitation, air temperature, CO_2_ level, or soil pH [33,34,35,36,37], while some studies deal with only one specific group of factors, most commonly edaphic [38,39,40], climatic [32,41] or habitat factors [42,43,44]. In addition, there is a difference in terms of analyzed data as well as examined mycocenoses. The fewest number of research papers deal with the influence of abiotic factors on the complete mycodiversity of an area, more precisely on the total number of fungi, the presence of different functional groups within the community, and the composition of species within communities [10,25,31,45,46,47]. Most studies have examined the influence of climatic factors on the production of fruiting bodies and the phenology of macrofungi [32,48,49], and the diversity of exclusively mycorrhizal or lignocolous fungi [14,30,50,51,52,53,54]. Concerning the mycorrhizal group, in most cases the influence of edaphic factors is examined, primarily the pH value, structure, content of mineral, and organic matter in the soil [55,56,57,58,59]. The ecological research of the lignicolous group was mostly related to habitat factors and characteristics of the wood substrate [34,42,60,61,62,63,64,65].

Compared with many other European countries, studies on the fungal diversity, ecology, and community dynamics in the specific forest habitats are still scarce in the region of the Mediterranean and Balkans, with Serbia representing one of the countries with the fewest published data. Therefore, the aim of this paper was to present the first results of the long-term monitoring of macrofungi in some of the most important mountain forest ecosystems in Serbia and to provide insight into their relationship with some of the abiotic factors (air temperature, air humidity, soil humidity, and precipitation) that are rapidly changing as a consequence of accelerated climate change and may lead to significant changes in fungal communities.

## 2. Materials and Methods

### 2.1. Study Sites and Experimental Plots

This research was carried out in three protected mountain areas in Serbia (Figure 1): Mitrovac, Mt. Tara (2 study sites); Metođe, Mt. Kopaonik (1 study site); and Vzganica, Mt. Vidlič (2 study sites). For long-term monitoring purposes, permanent experimental areas were specifically designed according to the standards set for the analysis of mycodiversity in forest ecosystems [66]. In each of the five examined forest sites, one rectangular or square experimental plot (1000 m^2^ in size) was marked. Each plot was chosen as a representative, homogeneous sample of the studied forest community.

Vidlič is predominantly positioned in the southeastern part of Serbia. The locality of Vzganica is located on the northeastern side of the Vidlič Mountain, at an altitude of 900 to 1200 m. It is characterized by a limestone substrate from the Jurassic period, which is covered with brown soil. Two experimental plots (“Plot 1” and “Plot 2”, Figure 1) were established in different forest types, both in an area under the III degree of protection, which belongs to the protected zone of the Stara Planina Nature Park [67]. Plot 1 was set in the stand of autochthonous beech, which regenerated independently after the fire in 1962 (*Fagus moesiaca* (K. Maly) Czecz.; central point 43°10′42.27″ N, 22°42′54.01″ E; altitude 1015 m above sea level; NE exposure; 6–15° slope). Plot 2 was set in the stand of a non-native species-Douglas fir, planted in the natural beech habitat after the aforementioned fire (*Pseudotsuga menziesii* (Mirb.) Franco; central point 43°10′51.17″ N, 22°42′31.05″ E; altitude 1015 m a.s.l.; NE exposure; ~10° slope).

Kopaonik represents the largest mountain range in the central part of Serbia, with the highest peak at 2017 m. The Metođe locality is a Nature Reserve, under the 1st degree of protection, within the Kopaonik National Park [67]. The specific geological base is in the form of granite, metamorphic rocks, and serpentine. On the specific part of the locality, where this research was carried out, there is a forest type of spruce with a beak-*Luzula sylvatica* (*Picetum-excelsea serbicum luzuletosom*). The experimental area, “plot 3”, is located on the northern slope of one of the high ridges that surround the valley of the stream that flows through Metođe (Figure 1). This forest stand is dominated by spruce trees of different ages (*Picea abies* (L.), with a smaller number of beech trees (*Fagus sylvatica* L.) mostly present in the young developmental stages (GPS coordinates: 43°18′18.1″ N, 20°50′38.4″ E; 1450 m a.s.l.; NE exposure; 30% slope).

Mountain Tara represents a part of the Dinaric Alps, located in western Serbia (the highest point-1591 m). Tara is one of the most important refugial habitats in Europe, with many rare, relict, and endemic species. Site Mitrovac is a large plateau located in the central part of Tara Mountain, at about 1080 m above sea level. It is characterized by the presence of a forest association of spruce, fir, and beech-*Piceo-Abietetum* Čol. 1965 (syn. *Piceo-Abieti-Fagetum moesiacum* Mišić et al., 1978), which is the most represented phytocenosis on Mt. Tara. Leaching from the surrounding higher areas and the accumulation of organic material in the Mitrovac valley resulted in a deeper, weakly acidic to basic soil, which represents a deposit of humus and undecayed parts of the forest litter. Two experimental areas in Mitrovac—“plot 4” and “plot 5”—represent stands of spruce (*Picea abies* (L.) Karsten), fir (*Abies alba* Mill.), and beech (*Fagus moesiaca* (K. Maly) Czecz) (Figure 1). Plot 4 is placed within the area under the II degree of protection (GPS coordinates: 43°55′06.1″ N, 19°25′33.6″ E; 1080 m a.s.l.; NE exposure; 3–5° slope). It is located on a flat plateau, bordered by streams and sinkholes on two adjacent sides, while the other two sides are bordered by parts of the forest that have been thinned by sanitary logging. Plot 5 (GPS coordinates: 43°55′00.18″ N, 19°25′11.61″ E; 1080 m a.s.l.; NE exposure; 3–5° slope) belongs to the peripheral part of the Nature Reserve “Crveni potok” (I degree of protection). There was no intensive exploitation of wood here, so thanks to this, the rainforest type of forest was preserved.

### 2.2. Fungal Sampling, Identification, and Data Analysis

Investigation of macrofungal diversity was conducted during the vegetation period of three consecutive years: 2011–2013 (four field trips during the first and last year and three field trips during the second year). Established permanent plots were carefully examined for the presence of fungal fruiting bodies. All macrofungal sporocarps were recorded, having in mind a simple presence-absence evaluation. Each fungus was photographed in the field, using a Nikon Coolpix P90 camera. A long period of transport and often unfavorable conditions (high temperatures, sporocarps damaged before sampling) resulted in the loss of one portion of collected samples or the possibility of identification only to the genus level. A laboratory examination implied an analysis of the macro- and micromorphological characteristics of fresh sporocarps, as well as specific chemical reactions. An examination of microscopic features was carried out on Olympus BX51, Japan. The Department of Literature’s Library, online books, keys, and specialized mycological sites were consulted during the identification process [68,69,70,71,72,73,74,75,76,77,78,79]. The nomenclature of the species names is in accordance with the database “Index Fungorum” (www.indexfungorum.org). Finally, representative specimens of each species were oven-dried on 50 °C (Scholtes FP 955.3). Dried material, spore prints, and microscope slides were deposited in the Fungarium collection of the ProFungi laboratory and within BUNS Herbarium (Department of Biology and Ecology, Faculty of Sciences, University of Novi Sad).

Species richness and composition served as the measures of macrofungal diversity. In order to assess the functional values of investigated sites, each fungal species was assigned to a specific functional–trophic group (lignicolus/terricolous saprotrophs/mycorrhizal) based on their primary mode of nutrition. The group of lignicolous fungi includes saprotrophs and parasites, given that only one species was found to be parasitic, but it was also found to be predominantlysaprotrophic at the same locality (*Fomitopsis pinicola* at the site Mitrovac, Tara Mt., plot 5). The group of terriculous saprotrophs also includes species that appeared within the litter, on the remains of leaves, needles, and twigs, due to the small number of such findings.

### 2.3. Measurement and Analysis of Abiotic Factors

In order to monitor microclimate characteristics, several parameters were measured at each site: temperature (T), relative air humidity (AH), and the dynamics of soil moisture (SM). Relative air humidity and air temperature in the stands were continuosly monitored by sensors, on hourly basis. The sensors were placed on trees, 2 m above ground. Based on these data, an analysis of mean daily and monthly temperatures was carried out. Soil moisture was determined at a depth of 10 cm. Samples were taken once a month from the same stand, at approximately the same place. Data on the precipitation (P) amount were obtained from the Republic Hydrometeorological Service of Serbia and represent the results from the weather stations nearest to each investigated site: “Zlatibor” (nearest to Mitrovac, Tara), “Kopaonik” (nearest to Metođe, Kopaonik), and “Dimitrovgrad” (nearest to Vzganica, Vidlič).

### 2.4. Statistical Analysis

The Sorensen similarity index was used in order to compare the investigated forest stands based on the species composition of their mycocenoses (i.e., the contribution of mutually similar macrofungal taxa in the total number of taxa of specific habitat pairs).

The obtained data on the species richness, distribution, and influence of abiotic factors were statistically processed using Microsoft Office Excel 2007 and XlStat Basic by Addinsoft (New York, NY, USA, www.xlstat.com, accessed on: 15 July 2022.). The following statistical analysis were applied: correlation analysis, correspondent analysis (CA), partial least square regression (PLS), and canonical correspondence analysis (CCA).

In order to examine the preference of recorded macrofungal species for certain types of forest habitats and their distribution, three-year data on their occurrence and representation within the examined experimental plots were subjected to multivariate statistical processing in the form of correspondent analysis (CA). For the purpose of the easier visualization of a large number of data, three separate analyzes of the identified species were performed, with one CA for each established functional group of macrofungi (mycorrhizal, lignicolous, and terricolous saprotrophs).

The influence of abiotic factors on the number of macrofungi within the mycocenoses of the observed forest habitats was determined by statistical processing of the obtained three-year results using the method of partial projections of the smallest squares, i.e., PLS (partial least square) regression.

In order to examine the redistribution of recorded macrofungal species within different habitats, according to abiotic environmental factors, the obtained three-year results were subjected to multivariate statistical processing through canonical correspondent analysis (CCA). This enables the simultaneous processing of two sets of different data (variables)—in this case, data on the frequency of species on certain plots and data on abiotic factors measured on given plots (T, P, AH, and SM).

## 3. Results and Discussion

### 3.1. Species Diversity and Representation of Macrofungi within Studied Forest Habitats

During the three-year study of five permanent plots, in different forest habitats, a total of 245 fungal taxa were recorded (Appendix A), belonging to the phyla Basidiomycota (227 taxa, 93%) and Ascomycota (18 taxa, 7%) and further classified within 100 genera, 53 families, 16 orders, and 5 class. The most represented genera were Mycena (43) and Russula (34), whose representatives make up 31% of the total diversity. These two genera appeared among the most represented ones in several other mycocenological papers [80,81,82,83,84]. The minority of the recorded taxa (10%) were present during each year of research. Species with a high frequency of occurrence during all three years were: *Cerioporus varius*, *Mycetinis alliaceus*, *Ganoderma applanatum*, *Hymenopellis radicata*, *Fomitopsis pinicola*, *Mycena pura*, and *Hypholoma fasciculare*.

Among the investigated mountainous forests, stands of spruce, fir, and beech in Mitrovac, Tara Mt. (Plot 4 and 5) stood out in terms of their macrofungal diversity (Figure 2). These well preserved old forests with diverse composition of woody species and water-rich substrates provide a good foundation for high fungal diversity. Several mycological studies in Central Europe have also shown that old mixed forests of beech, spruce, and fir are habitats of extremely diverse macrofungal communities, with a large number of rare and endangered species [85,86,87]. The protected forest habitats at Tara Mt. stand out, not only in terms of the total number of recorded species but also the presence of species that are interesting from a conservational perspective. Among them are species specialized in mosses as a specific substrate: *Galerina hypnorum*, *Rickenella fibula*, *Rickenella swartzii*, and *Entocybe nitida*, which are recognized as potential indicators of wet forest habitats [88,89,90]. *Hericium coralloide**s* and *Mycena laevigata* found at plot 5 are indicator species of valuable old forests [87,91,92] and together with the other values of this peripheral part of the Nature Reserve Crveni potok may serve for the consideration of the expansion of a strictly protected area.

According to the total number of findings and the identified macrofungal taxa, the stand of autochthonous Moesian beech on the site Vzganica, Vidlič Mt. (P1) and the stand of spruce and beech at Metođe, Kopaonik Mt. (P3) follow the habitats on Tara (Figure 2). The dominant fungal species in the indigenous beech forest (P1) were *Cerioporus varius*, *Hymenopellis radicata*, *Marasmius bulliardii*, *Megacollybia platyphylla,* and *Coprinellus xanthotrics*, respectfully, as well as *Phallus impudicus* and *Trametes versicolor* as the only species present on both experimental plots at Vzganica. These species seem to favor beech forests, which is in agreement with previous studies [26,80,86,93,94]. Important findings—unique for the autochthonous beech habitat on Vidlič (P1)—are *Flammulaster muricatus* and *Polyporus arcularius. F. muricatus* was evaluated as an indicator of valuable beech forests of special conservation importance at the European level [95], while both species were proposed as indicators of nature value in German forests [96].

The mixed forest stand on Kopaonik (P3) was also dominated by *C. varius* as the only species present during each month of research and always with a large number of sporocarps. This high frequency of sporocarps, covering a large area of examined sites, was also noted in the work of Simmel [90]. The other dominant species in this habitat, present several times during the three-year research, was *Amanita battarrae*, which was also recorded on P4 at Mitrovac (Tara). This mycorrhizal species is indigenous to Europe and is considered relatively rare [78,79].

Planted Douglas fir stand on Vzganica, Vidlič (P2) was singled out as the poorest macrofungal habitat (Figure 2), which was expected considering its altered nature. Among the investigated sites, this is the only planted forest stand with an introduced tree species—*Pseudotsuga menziesii* (Mirb.) Franco, originating from North America. Some of the studies in Europe determined that the number of macrofungi in the protected parts of forest reserves is up to 2 × higher in relation to the number of species in forests with active forest management, as well as in coniferous plantations in a beech habitat [26,87]. Other studies have also showed the decline of macrofungal species related to the conversion of forest habitats [81,97]. Among the macrofungi observed at allochthonous coniferous forest in Vzganica (P2), only the species *Amanita fulva*, *Amanita rubescens*, *Hydnum repandum*, *Cantharellus cibarius*, *Mycena galopus*, *Phallus impudicus*, *Laccaria laccata*, and *Hypholoma capnoides* were previously found to occur with Douglas fir [23,98,99]. Other species, recorded only at this forest stand (35 species, 73%), are not specific for Douglas fir as a partner or host.

### 3.2. Functional Diversity of Macrofungi

Fungal taxa identified in this study were recognized as the members of the following functional groups: lignicolous (104 species, 43%) > mycorrhizal (79 species, 32%) > terricolous saprotrophs (62 species, 25%). A similar representation of these functional groups was documented in other studies of forest macrofungal diversity [84,100].

The mycocenosis of spruce and beech forest at Metođe, Kopaonik Mt. (P3) had a similar representation of species within all three functional groups (Figure 3): 19 mycorrhizal species (32%), 22 lignicolous (37%), and 18 tericolous (31%). A similar result was noted within allochthonous Douglas fir stand at Vidlič (P2).

In terms of the number of species, the lignicolous group predominates among all of the studied plots, except the Douglas firs stand on Vzganica, Vidlič—plot P2. Mixed forests of spruce, beech, and fir on Mitrovac, Tara Mt. had the highest number of lignicolous fungi among the observed habitats (61 on P4, 35 on P5). Experimental areas on Tara are located in protected parts of the National Park with minimal interventions (the felling of trees, the extraction of fallen logs and stumps), which resulted in higher amount of wood residues suitable for the development of lignicolous fungi. Other studies of lignicolous fungi also showed higher species richness in study sites with a higher number of tree species and less human intervention [42,43,87,97,101,102,103].

Forest stands in Mitrovac, Tara Mt stood out with the highest number of mycorrhizal species (30 species on P4 and 21 species on P5), while the fungal community in Metođe, Kopaonik Mt. had the biggest share of this functional group. The smallest share of mycorrhizal macrofungi was observed in the stand of beech, P1, on Vidlič (6 species, 11%). These results are in agreement with the literature data showing that diversity of mycorrhizal species is positively associated with tree species diversity [104,105]. Additionally, based on the observations of several authors [93,106], the thickness of the litter and humus in Beech forests can affect the weaker development of mycorrhizal species. On the contrary, a relatively high share of the mycorrhizal group was found in the allochthonous forest habitat on Vidlič (P2-13 species, 27%). However, the mycorrhizal species found on P2 is typical for autochthonous deciduous and coniferous forests of this region. Among those, *Russula grisea* and *Lactarius vellereus* often prefer beech as a partner [107,108,109]. Similarly, *Lactarius volemus* and *Amanita fulva* were often reported within forest habitats of beech and spruce [78,107]. Thus, it may be questioned if the mycelium of the mycorrhizal species within the Douglas fir stand may originate from the surrounding indigenous beech forest or from spruces that also appear on the peripheral parts of Douglas fir plantation. A study by Jonsson et al. [110] showed that ectomycorrhizal communities of macrofungi have a high degree of continuity along a number of plots within a particular forest habitat.

Despite differences in the representation of the terricolous saprotrophs within each individual forest habitat, the number of recorded species of this functional group was uniform among different experimental plots. In the work of Dvořák et al. [87], a uniform diversity of terricolous saprotrophs in mixed stands with different management practices (without or with different degrees of forestry interventions) was found.

### 3.3. Distribution of Macrofungi across Studied Forest Habitats

Concerning species assemblages of studied forest habitats, the greatest similarity (Table 1) was observed amongst macrofungal communities of autochthonous mixed, predominantly coniferous, forest habitats (experimental plots P3, P4 and P5 (*Si*: 0.21–0.34)), while the broadleaf forest stand had lowest similarity with most of the other investigated plots (*Si*: 0.18–0.23; *Si*_sr_ (P1): 0.22). These data confirm the existence of differences in the composition of mycocenoses of deciduous and coniferous forest habitats, which has been noted in studies by other authors [80,111,112,113]. It was important to observe that beech forest stand on Vzganica, Vidlič Mt. (P1) showed the highest number of similar fungal species with the Douglas fir stand (P2) on the same locality.

Each of the three CA diagrams (Figure 4) represents a distribution biplot of identified species within a specific functional group and corresponding experimental plots. The group of mycorrhizal macrofungi had the largest share (90%) of species unique for only one of the investigated plots (presented as “Groups P1–P5” in the centrioles that overlap with the points of the plots where they were found), which is partly related to their host specificity [83,100,114]. Only eight mycorrhizal species (10%) were recorded within several investigated forest habitats, being mostly shared by coniferous and/or mixed forest stands: *Hydnum repandum* (P2, P5), *Amanita rubescens* (P2, P5), *Laccaria laccata* (P2, P3), *Boletus luridus* (P4, P5), *Porphyrellus porphyrosporus* (P3, P5), *Amanita battarrae* (P3, P4), and *Russula cyanoxantha* (P1, P4, and P5). Among these, only *Cantharellus cibarius* was found in four out of five experimental plots, being absent only from the purely deciduous habitat P1 at Vzganica, Vidlič Mt. These results confirm the findings of other authors, which demonstrated that aforementioned species have a broad prevalence and a wide host range [76,79,100,112,115].

Similar to the group of mycorrhizal species, among the recorded lignocolous macrofungi most species (76%) were found only in one of the examined forest habitats. This coincides with the results of the research on lignicolous macrofungi at the European level, which showed distinct differentiation in species composition among study sites (even those with the same tree species), which was linked with variations in wood decay stage, climatic conditions, and management history [100,102,116]. Among lignocolous species that demonstrated the widest distribution across study sites were *Bjerkandera adusta* (P1, P3, P4, and P5), *Calocera viscosa* (P2, P3, P4, and P5), *Cerioporus varius* (P1, P2, P3, and P5), *Hymenopellis radicata* (P1, P2, P3, and P4), *Pluteus cervinus* (P1, P3, P4, and P5), and *Xylaria hypoxylon* (P1, P2, P4, and P5). Only two lignicolous species were present in all of the investigated forest communities and experimental plots—*Hypholoma fasciculare* and *Mycena galericulata*. These lignicolous species, which are present within multiple forest stands, are known as species that are not strictly substrate-specific and are often present in beech and spruce forests in Europe [70,100].

Compared to the previous functional groups, terricolous saprotrophs have a smaller share of species unique for only one of the studied plots (71%). Unlike mycorrhizal and lignocolous fungi, they are lower substrate or habitat management preferences [83,100]. A recent study in China also reported that the saprotrophic group had the highest number of co-occurring genera within six different forest regions [84]. In our work, terricolous species distinguished for their high degree of distribution were *Lycoperdon perlatum* (P2, P3, and P5), *Gymnopus dryophilus* (P1, P2, P3, and P5), *Gymnopus androsaceus* (P2, P3, P4, and P5), and *Mycena sanguinolenta* (P2, P3, P4, and P5). Only *Mycena pura* and *Mycena galericulata* were noted in all five experimental plots and had the widest distribution among all terricolous saprotrophs.

### 3.4. Influence of Abiotic Factors on Macrofungal Species Richness

The results of the PLS analysis (Figure 5) are depicting the variability between overall data related to the number of macrofungal species (the total number of registered species and the number of species within different ecological groups) and measured abiotic factors across all of the study sites. Several groups of data can be observed, influenced by a different combination of abiotic factors.

The first group, consisting of the total number of identified species (TNS) and the number of lignicolous species (NoLig), was positively influenced by air humidity, precipitation, and soil moisture. It has already been recorded that the total number of registered macrofungal species in forests increases with increasing precipitation and air humidity [36,84,101,117]. Studies conducted in tropical forests in Costa Rica and Peninsular Malaysia [24,97] found that the number of lignicolous macrofungi increases with the increase in precipitation, air humidity, and soil moisture, which is in accordance with our results. Similarly, studies in European forests [35,101,118] found that macroclimatic factors (air temperature, precipitation, air humidity, and soil humidity) significantly affect the number of lignicolous fungi.

Data on the number of mycorrhizal macrofungi (NoMyc) and terricolous saprotrophs (NoTer) were isolated on opposite sides of the PLS diagram. The number of mycorrhizal macrofungi depended almost entirely on precipitation, but it also showed a positive correlation with air humidity. This confirms the observations of Lagan et al. [45], which also showed a significant dependence of the mycorrhizal macrofungi species richness in relation to precipitation. The number of terricolous saprotrophs in our research was mostly influenced by soil moisture and to a lesser extent by the air temperature, while precipitation and air humidity did not have a significant impact. Contrary to our findings, the abundance of terricolous saprotrophic fungi in the tropical forests of Malaysia was determined mainly by relative air humidity, next to habitat type and substrate richness [97].

### 3.5. Influence of Abiotic Factors on the Composition and Distribution of Macrofungal Communities across Studied Forest Habitats

The results of the CCA analysis are presented in “triplot” diagrams (Figure 6 and Figure 7). The only CCA analysis that did not provide results refers to the group of terricolous saprotrophs, indicating that their composition and distribution across the investigated plots cannot be explained by the influence of the examined abiotic factors. This is in agreement with the results of a group of authors [46] who examined the influence of environmental factors on the distribution and composition of macrofungi in coniferous pine forests in China. Their research (NMDS analysis) did not establish a significant relationship between saprotrophic macrofungi and analyzed abiotic factors (air temperature, humidity, and soil temperature).

Comparing the results of CCA (Figure 6) and PLS (Figure 5) analysis, it can be seen that the examined abiotic factors affect the species richness of the mycorrhizal group of macrofungi, as well as the specific redistribution of mycorrhizal species between different forest habitats, in a similar way. Such observations have so far not been recorded in the available mycocenological studies [31,33,35,46]. Based on the length of the vectors representing the gradients of abiotic factors, the variability of the observed data (distribution of species) is mostly contributed by the average monthly precipitation. Soil moisture and air humidity are somewhat less important, while the contribution of temperature is very small in relation to other environmental factors. These results are in line with the results of research on the variability of mycorrhizal communities of Scots pine, which showed that precipitation and soil moisture significantly affect the composition of ectomycorrhizal communities [119]. Studies of changes in mycorrhizal communities along the experimental hydrological gradient have shown that different soil moisture can significantly affect changes in the composition of mycorrhizal communities [120]. Studies based on experimental heat treatments have shown that an elevated temperature can also affect changes in mycorrhizal macrofungal communities [121,122].

By projecting the points that denote different species of mycorrhizal macrofungi on the corresponding vectors of the CCA diagram, we obtain information on what abiotic factors favored their appearance on certain experimental plots. The species concentrated in the center of the diagram are omitted in the interpretation of the results as they correspond to the mean values of the examined factors. *R. cyanescens* (observed at P1, P4, and P5), as well as all the species marked in blue on the diagram, which are unique for the plots P4 and P5 (Tara Mt.), stood out as species prone to wet, colder habitats. Their fructification was mostly affected by high SM, as well as high AH and higher amounts of P. The species grouped in the second quadrant (+ part of the F1 axis and − part of the F2 axis) corresponded to moderate air temperature and moderate amounts of precipitation. Species *Amanita rubescens* and *Cantharellus cibarius,* which are positioned closest to the center of the diagram, stood out as species that prefer moderate values of all examined parameters. In the works of other authors, they were also recognized as mesophilic species [31,123,124]. The same was concluded for several other species: *Hydnum repandum, Russula foetens,* and *Lactarius vellereus*, which were, on the contrary, in our research singled out as distinctly thermophilic macrofungi. Together with other species from the third quadrant (part of the F1 axis and part of the F2 axis), they were recorded in dry periods, with higher T. All species distributed in the negative part of the vertical axis were detected in the period of low humidity, among which only *Cortinarius croceus* is already listed in the literature as a common species in dry habitats [125].

According to CCA analysis for the lignicolous species, precipitation does not contribute to their distribution and representation within different plots, which was expected for this often xerophilous group of fungi. In order to obtain an appropriate model of variability depending on the remaining three examined abiotic factors (T, AH, and SM), the influence of precipitation was excluded from the analysis using the partial CCA statistical method (Figure 7). The pCCA triplot data for lignicolous species of macrofungi explains the overall 100% variability in species distribution of this functional group. The horizontal−F1 axis carries as much as 97.89% of the total variability, with the greatest contribution of AH and T, respectively, including SM, which was inversely correlated with them. The occurrence of species grouped in the first, positive quadrant (+part of the F1 axis, +part of the F2 axis) correlated with high AH and T. Species grouped in the second quadrant (+part of the F1 axis, negative part of the F2 axis) were observed in periods with moderate values of the examined abiotic parameters, while the appearance of lignicolous species from the third, negative quadrant coincides with the periods of low AH and T and high SM. As in the case of the analysis of mycorrhizal macrofungi, the species concentrated in the center of the diagram were omitted in the interpretation of the results, which means that they preferred moderate (mean) values of the examined factors. The influence of analyzed abiotic factors on the community structure of wood-inhabiting fungi is rarely documented in the literature. Lignicolous fungi are known to differ significantly in terms of their microclimatic preferences [61,126,127,128]. The results of some authors [30,51] showed that the specific structuring of lignicolous fungal communities is influenced to a greater extent by habitat and substrate characteristics (sun exposure, quantity, type, diameter, and the degree of the decomposition of the available wood material) than climatic factors. The importance of T and AH for the community composition of this fungal group, obtained in our research, is in agreement with two other studies [118]. Although usually neglected in the diversity analysis of lignicolous fungi, SM might play an important role in terms of maintaining the humidity of wood substrates that are in contact with the soil [65,101].

## 4. Conclusions

Species richness, arising in the order P2 < P3 < P1 < P5, together with all the important indicator species are pointing to the great importance of these forest habitats and the need to further improve their conservation. All of the obtained results indicate that the diversity of macrofungi reflects the state of the forest habitat and that analyzed abiotic factors strongly affect not only their species richness but also their community structure and distribution, which can also influence the overall balance of forest ecosystems. Thus, the continuation of long-term monitoring is crucial in order to more precisely determine which groups/species of mycrofungi, and to what extent, will adapt to a rapidly changing climate.

## Figures and Tables

**Figure 1 jof-08-01074-f001:**
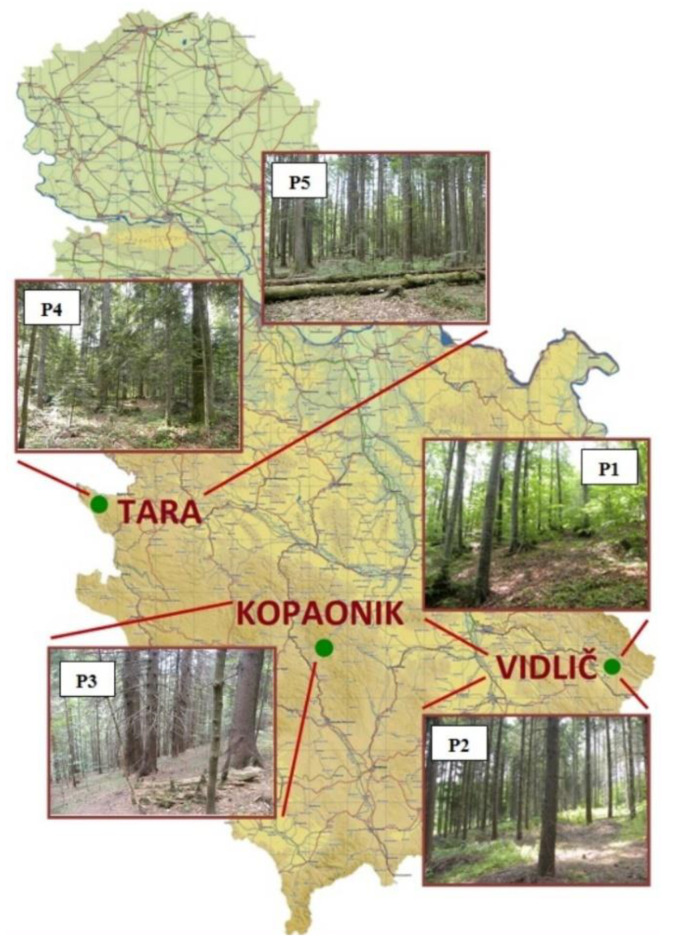
Map of Serbia, with locations of the study areas and representative forest plots.

**Figure 2 jof-08-01074-f002:**
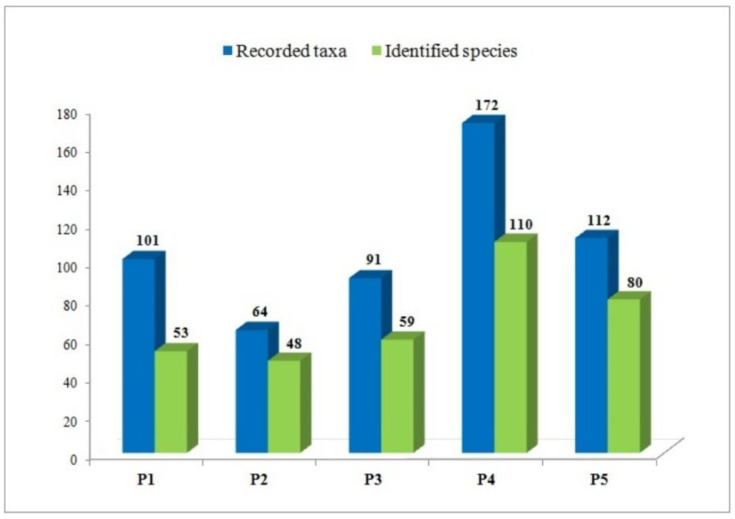
Map of Serbia, with locations of the study areas and representative forest plots.

**Figure 3 jof-08-01074-f003:**
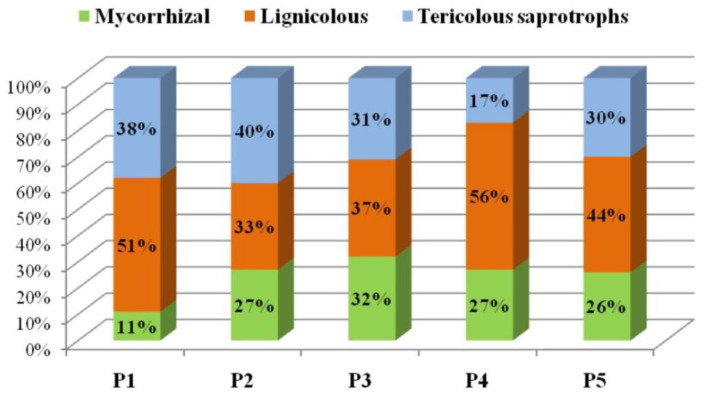
Functional diversity of macrofungi within different forest habitats.

**Figure 4 jof-08-01074-f004:**
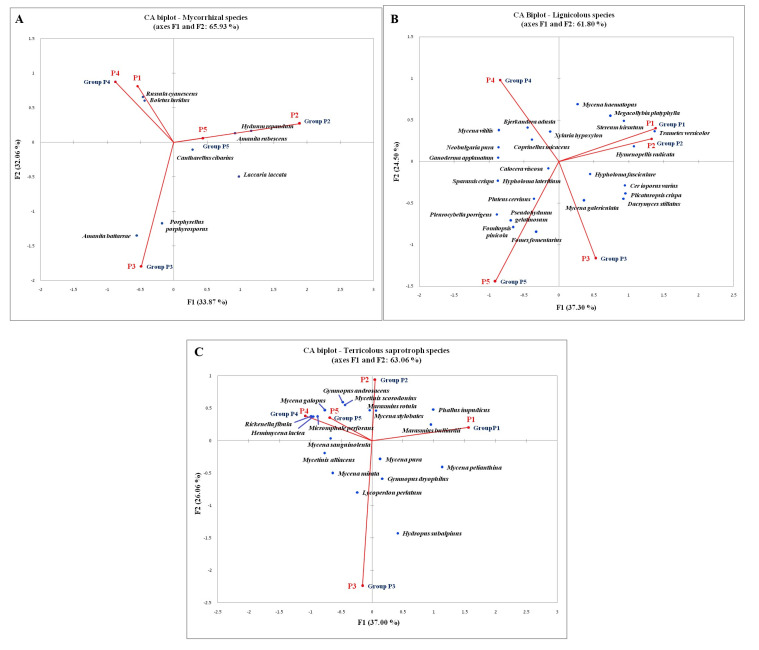
CA analysis of species distribution across studied forest habitats: (**A**) within the mycorrhizal group, (**B**) within the lignocolous group, and (**C**) within the group of terricolous saprotrophs. Legend: dots labeled as Group P1, P2, P3, P4, and P5 represent all the species found only within one specific plot (they are grouped in centrioles that overlap with the points of the corresponding plots).

**Figure 5 jof-08-01074-f005:**
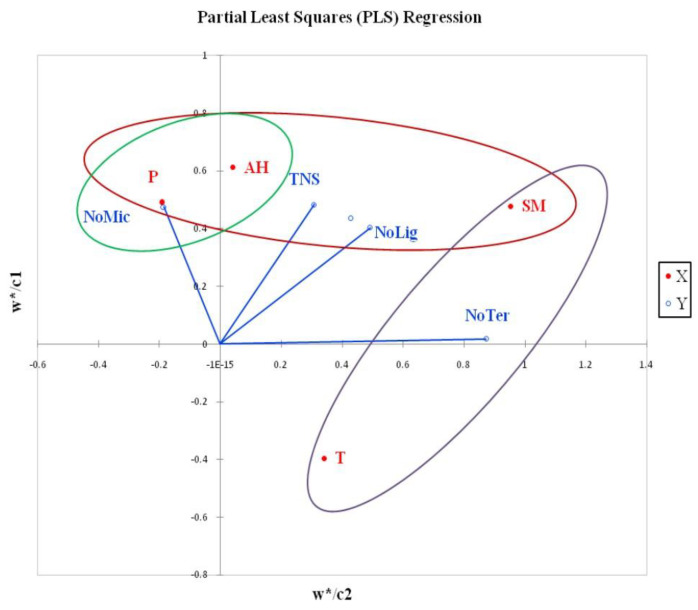
PLS analysis of the dependence of the macrofungal species richness on abiotic factors. Legend: independent variables (axis X): T−air temperature, P−mean monthly precipitation, AH−air humidity, SM−soil moisture; dependent variables (axis Y): TNS−total number of identified species, NoMik−number of mycorrhizal species, NoLig−number of lignicolous species, and NoTer−number of terricolous saprotrophs.

**Figure 6 jof-08-01074-f006:**
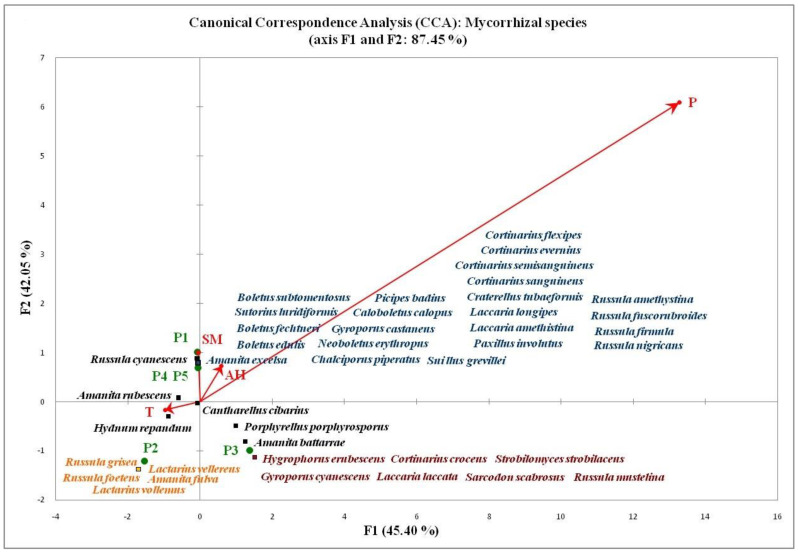
CCA analysis of the influence of abiotic factors on the composition of mycorrhizal group of macrofungi.

**Figure 7 jof-08-01074-f007:**
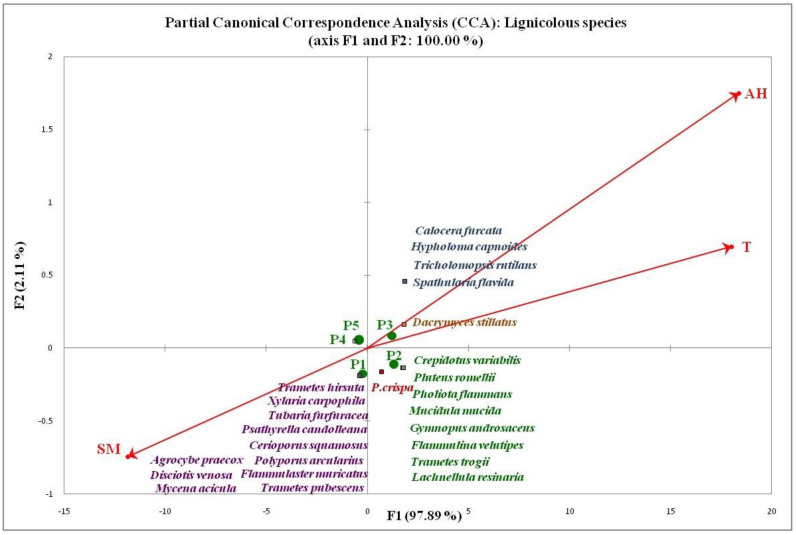
Partial CCA analysis of the influence of abiotic factors on the composition of lignocolous group of macrofungi.

**Table 1 jof-08-01074-t001:** Sorensen similarity index (Si).

	P1	P2	P3	P4	P5	*Si_av_*
P1		0.26	0.20	0.18	0.23	0.22
P2	*13*		0.24	0.17	0.30	0.24
P3	*11*	*13*		0.21	0.30	0.24
P4	*15*	*13*	*18*		0.34	0.23
P5	*15*	*19*	*21*	*32*		0.29

*Si**_av_*—average index value for each experimental plot; values in italic-number of macrofungi common to different forest habitats.

## Data Availability

All data obtained in this research are available upon request.

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
