# Peer review of "Diversity and Distribution of Macrofungi in Protected Mountain Forest Habitats in Serbia and Its Relation to Abiotic Factors"

_jof, 2022, doi:10.3390/jof8101074_

Round 1

Reviewer 1 Report

The manuscript in its current form cannot be published, although the results contained in it are quite interesting. A few comments to help the Authors rewrite the paper:

 Introduction - no citation of papers from the last 5 years;

 Material and Metods

No information in what years the mycological observations were conducted, only months are given. Here the question arises: why the Authors did not conduct research in spring (March-May) and autumn (October-November)?  It has long been known that many fungal species have an optimum of occurrence in late autumn. In addition, in southern Europe, climate changes, especially summer droughts, are not conducive to the appearance of macroscopic fungal fruiting bodies.  Thus, it should be explained why the authors conducted their research only in summer.

 The description of the study area is too long, a lot of information is unnecessary, such as lines 139-142:  „…two forest peatlands with a specific plant community - Crvena bara (Crveni potok) and Kurtina bara”.

 The Authors should focus on the description of the study plots, and in the supplementary materials include data on soil moisture, temperatures, etc., that were recorded on the study plots.

 As for the identification of fungi, it is necessary to state which literature items and determination keys were used. There is no information after whom the species names of fungi were given, Index Fungorum, MycoBank?

Results and Discussion

This part of the paper needs a major revision, first of all, it should be shortened, since the authors focused mainly on describing the results. They should be presented in a clearer and clearer way. In general, there is a lack of comparison of the obtained results with the results of other Authors - there are only a few references.

In addition, some of the conclusions included at the end of some parts (paragraphs) are conclusions not based on the observations made, e.g. „Species with the highest distribution (present at all investigated Mountains) were: Bjerkandera adusta, Calocera viscosa, Cantharellus cibarius, Cerioporus varius, Gymnopus androsaceus, Gymnopus dryophilus, Hymenopellis radicata, Lycoperdon perlatum, Mycena sanguinolenta, Pluteus cervinus, with the lead of Hypholoma fascisulare, Mycena galericulata and Mycena pura which were recorded at all 5 experimental plots. Such findings indicate a greater adaptability of these species to different habitats.”   The appearance of fungal fruiting bodies on the plot does not prove their adaptability. There are more such "conclusions" in the manuscript.

 I do not understand why the authors included Table 1. List of prominent species of researched habitats with data on endangerment in Serbia and European countries in the manuscript. Most of the fungal species included in this table are common in other countries, and are not on the red lists of fungi, e.g. Lactarius vellereus, Mycetinis scorodonius, and are not protected in Serbia. There are also errors in threat categories. In addition, there is no citation of the authors of the red lists for each country.

Conclusion

As it stands, it is a summary and conclusions. The text should be shortened and only 2-3 short conclusions should be given from the research.

Reviewer 2 Report

Dear authors, 

This paper is focusing on the Diversity and Distribution of Macrofungi in Protected Mountain Forest Habitats in Serbia and its relation to Abiotic Factors, which presents the results of a first comprehen-14 sive, long-term study of macrofungal communities in some of the most important mountain forest 15 ecosystems in Serbia (Tara, Kopaonik and Vidlič). It can be accepted after minor revison.

Please check the comments within the attached text. 

Kind Regards,

Round 2

Reviewer 1 Report

Dear Authors,

Please check the figure numbers in the text and figure captions.